# Study on a New Type of Composite Powder Explosion Inhibitor Used to Suppress Underground Coal Dust Explosion

Bo Liu [1,2], Yuyuan Zhang [1,2], Kaili Xu [1,2,*], Yansong Zhang [3,*], Zheng Hao [1,2] and Ning Ma [4]

1   College of Resources and Civil Engineering, Northeastern University, Shenyang 110819, China; skdliubo@126.com (B.L.); hellenzyy@163.com (Y.Z.); 18240138430@163.com (Z.H.)
2   Key Laboratory of Ministry of Education on Safe Mining of Deep Metal Mines, NO.3–11, Wenhua Road, Heping District, Shenyang 110819, China
3   College Safety and Environmental Engineering, Shandong University of Science and Technology, Qingdao 266590, China
4   Shandong Dingan Testing Technology Co., Ltd., Building 9, Area D, Era Headquarters Base, No.15 Lanxiang Road, Tianqiao District, Jinan 250000, China; mql7568@126.com
*   Correspondence: xukailineu@163.com (K.X.); zys6407@163.com (Y.Z.)

**Abstract:** At present, the world is committed to the development of environmentally friendly, sustainable and industrial safety. The effective treatment of industrial solid waste can be applied in the field of industrial safety. It is one of the ways to apply industrial solid waste to industrial safety to modify industrial solid waste and combine active powder to prepare industrial solid waste-based composite powder explosion inhibitors and apply it to underground coal dust explosion. This paper introduces the modification and preparation methods of industrial solid waste, and analyzes the good explosion suppression effect and good economic benefit of industrial solid waste-based composite powder explosion inhibitors on coal dust explosion. In this paper, four kinds of industrial solid wastes (red mud, slag, fly ash and sludge) were modified, and the modified solid waste materials with good carrier characteristics were obtained. Combined with a variety of active powders ($NaHCO_3$, $KH_2PO_4$ and $Al(OH)_3$), the industrial solid waste-based composite powder explosion inhibitors were obtained by solvent-crystallization (WCSC) and dry coating by ball milling (DCBM). Those kinds of explosion inhibitors can suppress the explosion of pulverized coal in 40–50% of cases. Compared with the powder explosion inhibitor commonly used in industry, it has a lower production cost and better explosion suppression effect. Those kinds of explosion inhibitors have a good industrial application prospect.

**Keywords:** industrial solid waste; inhibitor; modification; coal dust explosion; compounding technology



## 1. Introduction

Continued industrialization has brought along increased emissions of industrial solid wastes. Conventional land-based stockpiling and landfill not only contaminate land and water sources, but also compromise people's health. In the resource utilization of industrial solid wastes, few people have given it a consideration that industrial solid wastes can be used as a raw material to make new construction material or other materials [1–3]. Thus far, bulk industrial solid wastes are treated primarily by incineration and secondarily by solidification–landfill [4]. While incineration and solidification–landfill can deal with large volumes of industrial solid wastes, as incineration produces toxic and green gases, it leads to even worse air pollution problems; solidification–landfill is also harmful to groundwater resources by generating large amounts of leachate. In fact, most industrial solid wastes can be reused as resources through chemical, physical or biological modification to remove the toxic contents or neutralize the pH value [5–7].

Studies on the reuse of red mud resources are mostly concentrated on extracting ferro-aluminum contents or using it as a catalyst or to synthesize new materials. Li et al. [8]



tried to remove the $SO_2$ and $NO_x$ in industrial flue gas simultaneously using red mud as the absorbent together with $O_3$. Wang et al. [9] used dealuminized red mud and fly ash to prepare composite flocculant for treating diatomite simulated wastewater. The explosion inhibitor made from red mud is mostly used in the experimental study of gas explosion suppression. Yu et al. [10] modified red mud to obtain an ultrafine modified red mud powder explosion inhibitor, which has good performance in a gas explosion suppression experiment. Wang et al. [11] used urea and fly ash to synthesize a core-shell composite powder explosion inhibitor, which has better effect on a coal dust explosion than fly ash and urea. Cristelo et al. [12] used municipal solid waste incineration (MSWI) products—fly ash (FA) and bottom ash (BA)—together with sodium silicate or sodium hydroxide to prepare the three new types of slurry, respectively. Mechanical testing proved that all three pastes provide a mechanical property qualifying for engineering application. Long et al. [13] used MSWI-FA as the carrier of immobilization. It is feasible to cotreat the ferronickel slag (FNS) using geopolymer technology. Ref. [14] studied the feasibility of lime-stabilized sludge as a building material for a flexible pavement subbase. Through a comprehensive study of the literature on the utilization of industrial solid waste, it is found that there are relatively few studies on the use of industrial solid waste as an explosion inhibitor, and there are many applications in the prevention of gas explosion, and there is a lack of application research on coal dust explosion in underground coal mines.

The aforementioned four industrial solid wastes are ideal carrier materials for composite powder explosion inhibitors because of the following reasons [15,16]. First, the output of solid waste is large and the cost is low. Huge amounts of industrial solid wastes are generated every year and they are available at very cheap prices. Second, they contain explosion suppression components. The main components of the four solid wastes are $SiO_2$, $CaO$, $Al_2O_3$, $Fe_2O_3$, $MgO$ and $Na_2O$. Various components are inert powder and have the ability to suppress an explosion. In addition, the $Al_2O_3$ and $Fe_2O_3$ oxides contained in solid waste can be converted into metal hydroxide $Al(OH)_3$ and $Fe(OH)_3$ with certain explosion inhibition activity [17]. Third, they contain a lot of micropore structures. The abundant pore structures allow the industrial solid wastes to carry other active supporters and realize the synergistic explosion suppression effect of solid waste and active explosion suppression powder.

In this paper, four kinds of solid wastes—red mud, fly ash, slag and sludge—are studied, and a variety of new composite powder explosion inhibitors are prepared through a modification and compounding process. In view of underground coal dust, a new type of anti explosion agent is compared with the commonly used anti explosion agent ABC powder on the market, and the explosion suppression effect and cost analysis of various new composite powder inhibitors are studied. The feasibility of preparing an effective new composite powder explosion inhibitor from typical industrial solid waste was discussed. The preparation method of the new explosion inhibitors provides a new way for the resource of industrial solid waste to be utilized. The explosion suppression application of the new explosion inhibitors to coal dust provide a new method for the prevention of underground coal dust explosion accidents.

## 2. Materials and Methods

### 2.1. Modification of Industrial Solid Wastes

The red mud used for our experiment consisted of red mud waste from aluminum oxide production by Bayer process at Nanshan Aluminum General Plant of Nanshan Aluminum in Longkou, Shandong province. The slag and fly ash consisted of bottom slag from the main MSWI boilers and fly ash collected by bag filters at Hengyuan Thermal Power Waste Incineration Power Plant in Qingdao, Shandong province. The sludge consisted of sludge precipitated from acrylic production sewage treatment at the acrylic plant of PetroChina Daqing Petrochemical Co., Ltd in Daqing, Hei Longjiang province. For four kinds of solid wastes, XRF analysis was carried out using a Brucker AXS X-ray fluorescence analyzer (XRF, S8TIGER4kW), as shown in Table 1.

**Table 1.** XRF analysis of solid wastes.

| Substance | Fe$_2$O$_3$ | Al$_2$O$_3$ | SiO$_2$ | CaO | TiO$_2$ | Na$_2$O | MgO | Cl |
|---|---|---|---|---|---|---|---|---|
| Red mud | 33.3 | 18.2 | 15.3 | 16.3 | 6.73 | 8.19 | 0.30 | 0 |
| Slag | 12.7 | 26.3 | 41.2 | 7.17 | 6.73 | 1.20 | 2.35 | 3.22 |
| Fly ash | 16.923 | 13.834 | 2.69 | 11.7 | 1.22 | 5.6 | 1.88 | 22.6 |
| Sludge | 36.8 | 12.4 | 12.4 | 2.89 | 0.68 | 5.72 | 1.01 | 0 |

As indicated, all four solid wastes contain high levels of metal oxides and SiO$_2$. Hence, it is necessary to acidify–alkalize them to generate metal oxides useful for suppressing explosion. Slag and fly ash contain a level of element Cl since both contain some dioxin, a carcinogenic toxic substance; hence, they have to be detoxified [18]. The material analysis of sludge shows that the content of organic matter in sludge is 50.2%, PH value is 5.56 (acidic), total nitrogen is 12.4, total phosphorus is 0.152 and total potassium is 0.236. The sludge contains 50.2% of organic matter such as additives in the production process of acrylic fiber; hence, it should be treated with decombustion. According to the related literature [19–21], the dioxin in slag and fly ash can be fully decomposed by high-temperature melting, and high-temperature combustion is also needed to remove the burning materials from sludge. Hence, a Muffle roaster was used to combust the slag, fly ash and sludge under high temperature before a planetary ball mill was used to crush them to 75 μm to form the preliminary modified products.

Next, the red mud and the preliminarily modified slag, fly ash and sludge were acidified–alkalized in the following procedure (in the case of red mud as an example):

(1) Into 100 mL deionized water, 30 g CN was added. To this, 6 mol/L HCl was added dropwise, and the metal salt solution was obtained by stirring at 90 °C for 1.5 h with a magnetic stirrer.

(2) Ammonia water was added to the metal salt solution to the pH value of 7.8; then, 150 mL of Absolute ethanol (C$_2$H$_6$O) was added, and the mixture was stirred at 60 °C for 1 h.

(3) After the substance in the solution was precipitated, it was washed and filtered with non-ionic water using a vacuum pump. The filtered mud was dried in vacuum drying oven for 24 h, and the modified red mud (MRM) with particle size less than 75 μm was obtained by grinding with ball mill.

In the same way, modified slag (MSA), modified fly ash (MFA) and modified sludge (MSU) were obtained. Figure 1 compares the SEM results of the microstructure morphologies of the four industrial solid wastes before and after modification.

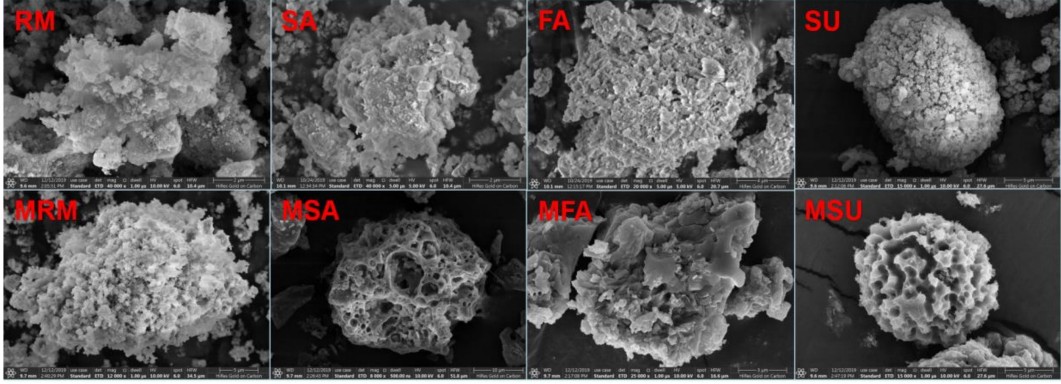

**Figure 1.** SEM analysis of four kinds of industrial solid wastes before and after modification.

As illustrated, after modification, MRM, MSA, MFA and MSU have become porous materials with a much higher porosity, which greatly enhanced the loading capacity of the materials.

### 2.2. Compounding Process of Industrial Solid-Waste Based Composite Powder Explosion Inhibitors

Active powder inhibitors $NaHCO_3$, $KH_2PO_4$ and $Al(OH)_3$ were used as the supporters of the new composite powder explosion inhibitors, mainly because of the following reasons [22,23]:

(1)  All three active powder inhibitors decompose thermally under heating conditions and absorb large amounts of heat in the explosive environment, thereby suppressing explosion by thermolysis endothermic cooling.

(2)  The $Al_2O_3$ from thermal decomposition of $Al(OH)_3$ can attach to exploding particles during explosion suppression, thereby isolating thermodynamic activity.

(3)  All three inhibitors can decompose thermally to generate gases or vapors such as $CO_2$, $P_2O_5$ and $H_2O$. The reaction product can dilute the volatile gases and oxygen, thereby inhibiting explosion.

(4)  All three inhibitors contain substances that can reduce the activity of free radicals (e.g., O, OH, HCO· and OHP·), which participate in the explosion reaction, thereby inhibiting the combustion–explosion reaction rate.

The four industrial solid wastes were composited with inert powder inhibitors $NaHCO_3$, $KH_2PO_4$ and $Al(OH)_3$ as the supporters of the new composite powder explosion inhibitors, using wet coating method of solution-crystallization (WCSC) and dry coating method using ball milling (DCBM) methods. As $Al(OH)_3$ does not dissolve in water or ethanol, DCBM method was used. The composite powder explosion inhibitors were prepared in the following procedure (in the case of slag-based composite powder explosion inhibitor as an example).

The prepared MSA was composited with the carrier into slag-based composite powder material. All carrier materials (except $Al(OH)_3$) selected were powders soluble in water but not soluble in ethanol. As MSA is not soluble in either water or ethanol, the WCSC method is used to prepare slag-based composite powder explosion inhibitor, which takes advantage of the solubility difference of the carrier in water and ethanol (see Table 2). The following procedure was used (in the case of MSA-$KH_2PO_4$ composite powder explosion inhibitor as an example) (see Figure 2a).

**Table 2.** Solubility of different solutes.

| Solvent<br>Solute | Deionized<br>Water | Ethanol | Decomposition<br>Temperature | Preparation Method | | |
|---|---|---|---|---|---|---|
| | | | | **WCSC** | **DCBM** | **DCAI** |
| $NaHCO_3$ | 9.6 | Insoluble | 50 °C | √ | √ | √ |
| $KH_2PO_4$ | 22.6 | Insoluble | 252.6 °C | √ | √ | √ |
| $Al(OH)_3$ | Insoluble | Insoluble | 298 °C | — | √ | √ |

Into 150 mL of $C_2H_6O$, 100 g of MSA was added and magnetically stirred to form a uniform suspension. Saturated solution was prepared according to the water solubility of the supporting component $KH_2PO_4$. Next, 22.6 g of $KH_2PO_4$ was weighed and dissolved in deionized water (at a constant water temperature of 25 °C). Saturated $KH_2PO_4$ solution was slowly added into the MSA-$C_2H_6O$ suspension at a uniform rate (at an experimental temperature of 25 °C and a constant speed of 700 r/min). As crystallization occurred due to sudden reduction in the solubility of $KH_2PO_4$ in the mixed solution, after the $KH_2PO_4$ solution was fully added, excess $C_2H_6O$ was added at the same speed until the $KH_2PO_4$ crystals were completely separated onto the surface of MSA and became precipitates. After ultrasonic dispersion for 30 min, the precipitate was filtered out, and then dried in vacuum drying oven for 24 h, thus MSA-$KH_2PO_4$ composite powder explosion inhibitor with particle size less than 75 was formed.

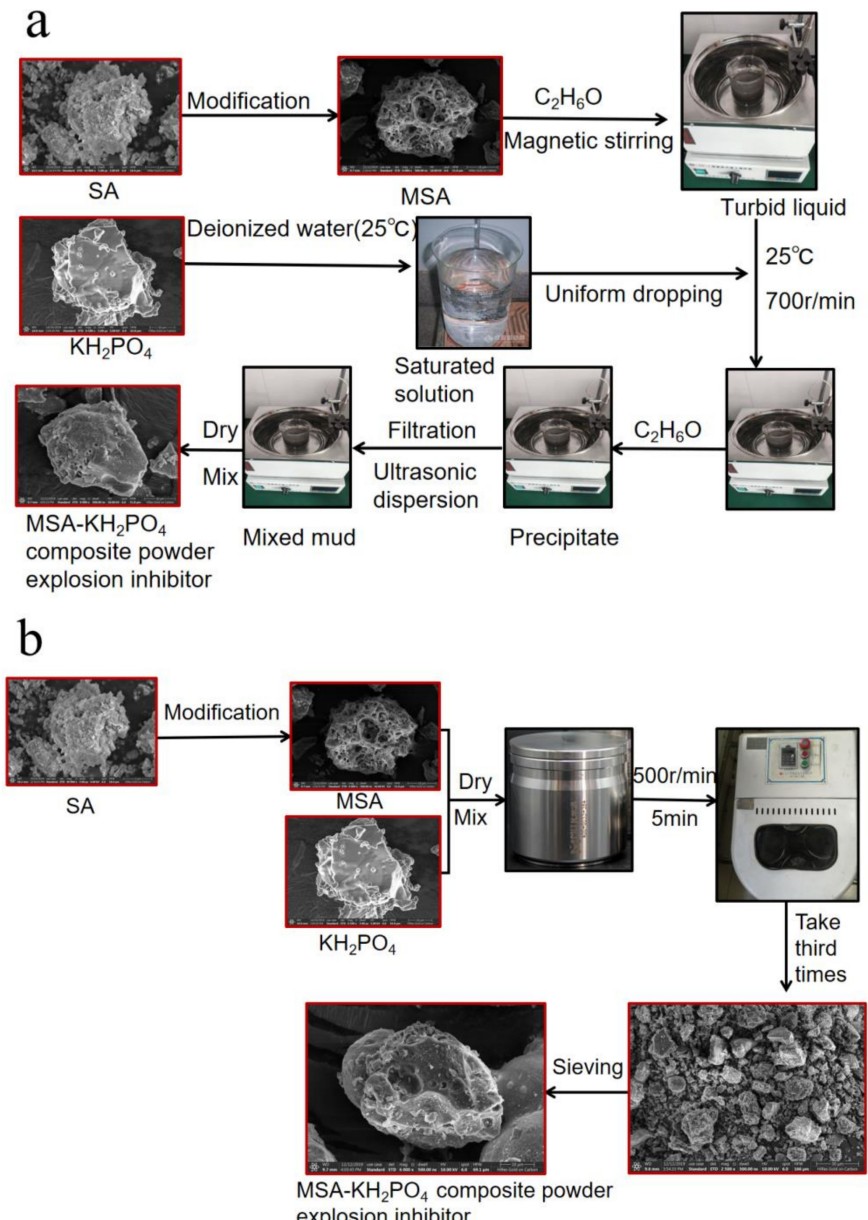

**Figure 2.** Experimental steps of WCSC method and DCBM method. (**a**). WCSC method; (**b**). DCBM method.

Slag-based composite powder explosion inhibitors were prepared using DCBM method in a TC-XQM4 vertical semicircular planetary ball mill (Changsha Tianchuang Powder Technology Co., Ltd., Changsha, China). During preparation, the two powder components were fully ground. Under friction and extrusion, the carrier was embedded into the pores of the supporter to form a coating on the supporter. Slag-based composite powder explosion inhibitors were prepared in the following procedure (in the case of MSA-$KH_2PO_4$ composite powder explosion inhibitor as an example) (see Figure 2b).

The powders were dried in a vacuum dryer under 30 °C for 24 h. Then, 100 g of the slag and 22.6 g of $Al(OH)_3$ were weighed, placed together into the steel tank and ground at 500 r/min for 5 min. Samples from the first and second operations were disposed as machine-washing waste. That from the third operation was used as the finished product.

MSA-$NaHCO_3$ composite powder explosion inhibitor, MSA-$KH_2PO_4$ composite powder explosion inhibitor and MSA-$Al(OH)_3$ composite powder explosion inhibitor were prepared using these two methods. To observe the microstructure morphologies and com-

positing degree of the samples under these two methods, scanning electron microscopy (SEM) and BET testing were conducted. To facilitate subsequent studies, the materials prepared using WCSC method are named WCSC-MSA-NaHCO$_3$ composite powder explosion inhibitor and WCSC-MSA-KH$_2$PO$_4$ composite powder explosion inhibitor. The material prepared using DCBM method is named DCBM-MSA-Al(OH)$_3$ composite powder explosion inhibitor.

A JSM-6390LV SEM was used to compare the microstructure morphologies of the composite inhibitors prepared using the two methods.

As illustrated, the modified slag is porous with developed pores and is, therefore, an ideal carrier material. KH$_2$PO$_4$ is irregularly and subsmooth in shape. In the image, after WCSC process, it is perfectly coated on the modified slag. In Figure 3, almost all slag pores are filled up. The coating is full and well tensioned. Obviously, the composite suppressor prepared using WCSC method is well composited, as expected. In Figure 3, after DBCM process, the slag pores are basically filled up, except for only a very small part. The coating is not so full, but the overall coating is good. The composite suppressor prepared using DCBM method is also well composited, as expected.

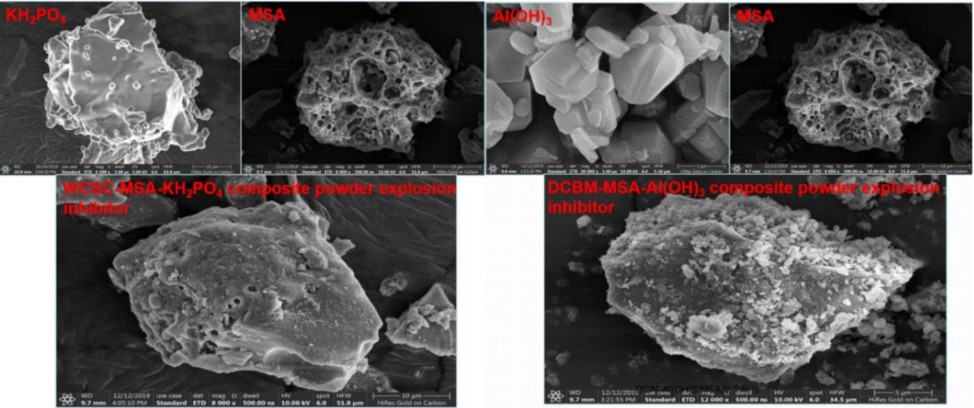

**Figure 3.** SEM analysis of four kinds of slag-based composite powder explosion inhibitors.

An Autosorb-i Q automatic BET and pore size analyzer were used to compare the microstructure morphologies of the composite inhibitors prepared using the two methods BET and porosity before and after preparation.

As shown in Table 3, BET and pore volume of slag-based composite powder explosion inhibitors are much lower than those of MSA, because NaHCO$_3$, KH$_2$PO$_4$ and Al(OH)$_3$ can be effectively adsorbed and coated on MSA after the compounding process. As illustrated by Figure 4, the nitrogen adsorption desorption isotherm curve of MSA shows a type IV-H2(a) hysteresis loop in IUPAC classification, which indicates that MSA has relatively uniform channels [24,25]. The isothermal curves of WCSC-MSA-NaHCO$_3$ composite powder explosion inhibitor and WCSC-MSA-KH$_2$PO$_4$ composite powder explosion inhibitor are type III. As indicated by Figure 3 (SEM), both composite powder explosion inhibitors have become nonporous materials. The isotherm of DCBM-MSA-Al(OH)$_3$ composite powder explosion inhibitor appears to be an IV-H3 hysteresis loop, which suggests pores with plate slit structure, fracture and wedge structure. An IV-H3 hysteresis loop is provided by schistose granular material such as clay, or by fissure pore material, with no adsorption saturation shown in the relative pressure region. This confirms that the composite powder explosion inhibitors prepared using WCSC and DCBM methods are very well composited.

**Table 3.** Analysis of BET and pore volume of composite powder explosion inhibitors.

| Sample | BET (m²/g) | Pore Volume (cm³/g) |
|---|---|---|
| MSA | 567.6321 | 0.416673 |
| WCSC-MSA-NaHCO₃ composite powder explosion inhibitor | 68.4097 | 0.198494 |
| WCSC-MSA-KH₂PO₄ composite powder explosion inhibitor | 55.9697 | 0.149462 |
| DCBM-MSA-Al(OH)₃ composite powder explosion inhibitor | 115.2649 | 0.166799 |

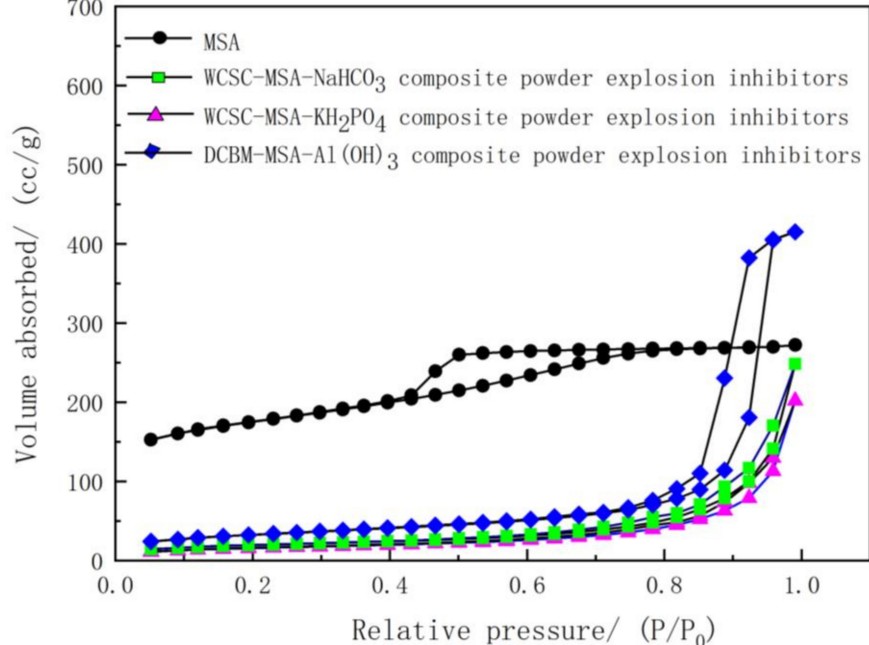

**Figure 4.** Nitrogen adsorption desorption results of industrial solid wastes-based composite powder explosion inhibitors.

Industrial solid waste-based $NaHCO_3$ composite powder explosion inhibitor, industrial solid waste-based $KH_2PO_4$ composite powder explosion inhibitor and industrial solid waste-based $Al(OH)_3$ composite powder explosion inhibitor were prepared using WCSC and DCBM method. Table 4 gives the process names and suppressor names of these composite powder explosion inhibitors.

From the changes in the BET values of the supporters and the industrial solid waste-based composite powder explosion inhibitors, we can see the compositing efficiency, namely, the load ratio, of the industrial solid waste-based composite powder explosion inhibitors as follows:

$$\eta = \frac{BET_{Supporter} - BET_{Industrialsolidwaste-basedcompositepowderexplosioninhibitor}}{BET_{Supporter}} \quad (1)$$

As indicated in Figure 5, the loading rate of modified solid waste materials on active explosion inhibitors ($NaHCO_3$, $KH_2PO_4$ and $Al(OH)_3$) is more than 85%. By BET and pore volume, the modified industrial solid wastes are sorted as MSA > MSU > MCN > MFA. After the modified solid wastes are loaded with $NaHCO_3$, $KH_2PO_4$ and $Al(OH)_3$, their BETs and pore volumes largely drop. WCSC and DCBM caused the supporters to coat the carriers effectively. Now by BET and pore volume, the industrial solid waste-based composite powder explosion inhibitors are sorted as slag-based composite powder explosion

inhibitor > red mud-based composite powder explosion inhibitor > sludge-based composite powder explosion inhibitor > fly ash-based composite powder explosion inhibitor.

**Table 4.** Name list.

| Carrier | Process Name | Suppressor Name |
|---|---|---|
| Modified red mud | Red mud-based NaHCO$_3$ composite powder explosion inhibitor | MCN-NaHCO$_3$ composite powder explosion inhibitor |
| | Red mud-based KH$_2$PO$_4$ composite powder explosion inhibitor | MCN-KH$_2$PO$_4$ composite powder explosion inhibitor |
| | Red mud-based Al(OH)$_3$ composite powder explosion inhibitor | MCN-Al(OH)$_3$ composite powder explosion inhibitor |
| Modified slag | Slag-based NaHCO$_3$ composite powder explosion inhibitor | MSA-NaHCO$_3$ composite powder explosion inhibitor |
| | Slag-based KH$_2$PO$_4$ composite powder explosion inhibitor | MSA-KH$_2$PO$_4$ composite powder explosion inhibitor |
| | Slag-based Al(OH)$_3$ composite powder explosion inhibitor | MSA-Al(OH)$_3$ composite powder explosion inhibitor |
| Modified fly ash | Fly ash-based NaHCO$_3$ composite powder explosion inhibitor | MFA-NaHCO$_3$ composite powder explosion inhibitor |
| | Fly ash-based KH$_2$PO$_4$ composite powder explosion inhibitor | MFA-KH$_2$PO$_4$ composite powder explosion inhibitor |
| | Fly ash-based NaHCO$_3$ composite powder explosion inhibitor | MFA-Al(OH)$_3$ composite powder explosion inhibitor |
| Modified sludge | Sludge-based NaHCO$_3$ composite powder explosion inhibitor | MSU-NaHCO$_3$ composite powder explosion inhibitor |
| | Sludge-based KH$_2$PO$_4$ composite powder explosion inhibitor | MSU-KH$_2$PO$_4$ composite powder explosion inhibitor |
| | Sludge-based NaHCO$_3$ composite powder explosion inhibitor | MSU-Al(OH)$_3$ composite powder explosion inhibitor |

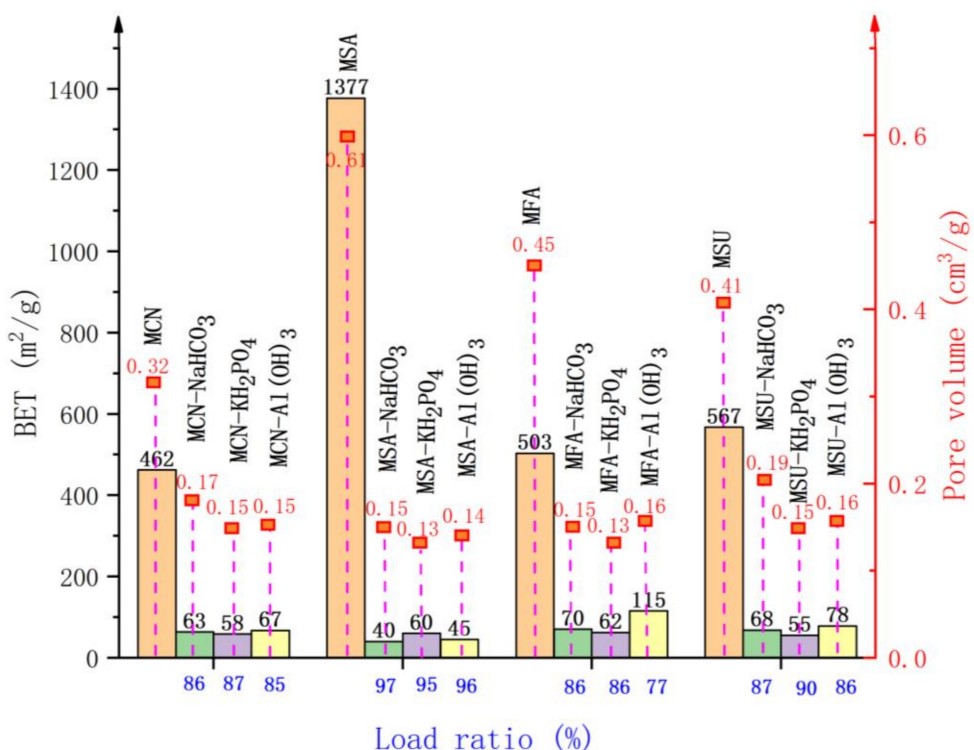

**Figure 5.** Analysis of BET and pore volume of composite powder explosion inhibitors.

## 3. Result and Discussion

### 3.1. Suppression Effect of Industrial Solid Waste-Based Composite Powder Explosion Inhibitors on Coal Dust Explosion

It is very important to study the characteristics of the explosion suppression of coal dust in an enclosed space for the explosion protection parameters of explosion protection facilities (such as the key parameters of venting area and venting pressure). The variation of the maximum explosion pressure and maximum explosion pressure rising rate can reflect the inhibition effect of explosion inhibitor on coal dust explosion. According to the standard GB/T16425, the standard 20 L spherical explosion system was used for the explosion suppression experiment [26–28], as indicated in Figure 6. The test apparatus is composed of the following three parts: the main spherical tank, the control system and the data acquisition system. The main spherical tank is made of stainless steel, and a nozzle and ignition lead are built in the main spherical tank. In the experiment, a given weight of dust was placed in the dust vessel. A 10 KJ chemical igniter mounted at the center was connected to the ignition lead. The explosion vessel was safely closed. The explosion chamber was vacuumed to 0.06 MPa. The dispersion gas pressure was set to 2.0 MPa. When the solenoid valve between the dust vessel and the test chamber started automatically, air and dust were injected into the explosion chamber and ignited after a 60-millisecond time delay. The mixture of coal dust with the $MCN-NaHCO_3$ composite powder explosion inhibitor, the $MCN-KH_2PO_4$ composite powder explosion inhibitor and the $MCN-Al(OH)_3$ composite powder explosion inhibitor was carried out. The explosion characteristics of the mixture of single powder (pure $NaHCO_3$, pure $KH_2PO_4$, pure $Al(OH)_3$ and pure MCN) and coal dust were compared. Into the coal dust, 0, 10, 20, 30, 40, 50, 60, 70, 80, 90 and 100% of the explosion suppression powder were added.

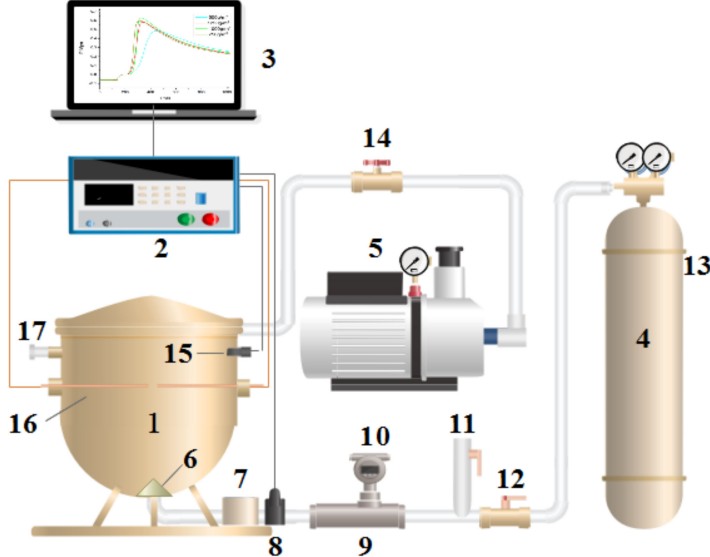

**Figure 6.** 20 L spherical explosion system. (1) 20-liter spherical tank, (2) Testing system for explosive characteristics electrode, (3) Computer, (4) High-pressure air cylinder, (5) Vacuum pump, (6) Nozzle, (7) Powder storage bin, (8) Solenoid valve, (9) High pressure air storage vessel, (10) Pressure gauge, (11) Exhaust valve 1, (12) Air inlet valve, (13) Pressure relief valve, (14) Vacuum pump ventilation valve, (15) Pressure sensor, (16) Ignition lead, (17) Exhaust valve 2.

Figure 7 shows the explosion pressure curve of the explosion inhibitor for suppressing a coal dust explosion. The results show that the maximum explosion pressure ($P$max) and maximum rate of explosion pressure rise (($dp/dt$)max) of the coal dust decrease under the action of an explosion inhibitor. A 40% $MCN-NaHCO_3$ composite powder explosion inhibitor, an $MCN-KH_2PO_4$ composite powder explosion inhibitor and a 50% $MCN-Al(OH)_3$ composite powder explosion inhibitor can completely inhibit the explosion of coal dust,

and the inhibition effect of single powder is not as good as that of a red mud-based composite powder explosion inhibitor. The $P$max of coal dust is 0.64 MPa, and the $(dp/dt)$max is 26.5 MPa/s. After the red mud composite powder explosion inhibitor is added, the maximum pressure reduction rate of the coal dust is 52.3% for a 30% MCN-NaHCO$_3$ composite powder explosion inhibitor, 49.9% for a 30% MCN-KH$_2$PO$_4$ composite powder explosion inhibitor and 46.7% for a 40% MCN-Al(OH)$_3$ composite powder explosion inhibitor. By coal dust explosion suppression efficiency, the composite powder explosion inhibitors are sorted as MCN-NaHCO$_3$ composite powder explosion inhibitor > MCN-KH$_2$PO$_4$ composite powder explosion inhibitor > MCN-Al(OH)$_3$ composite powder explosion inhibitor.

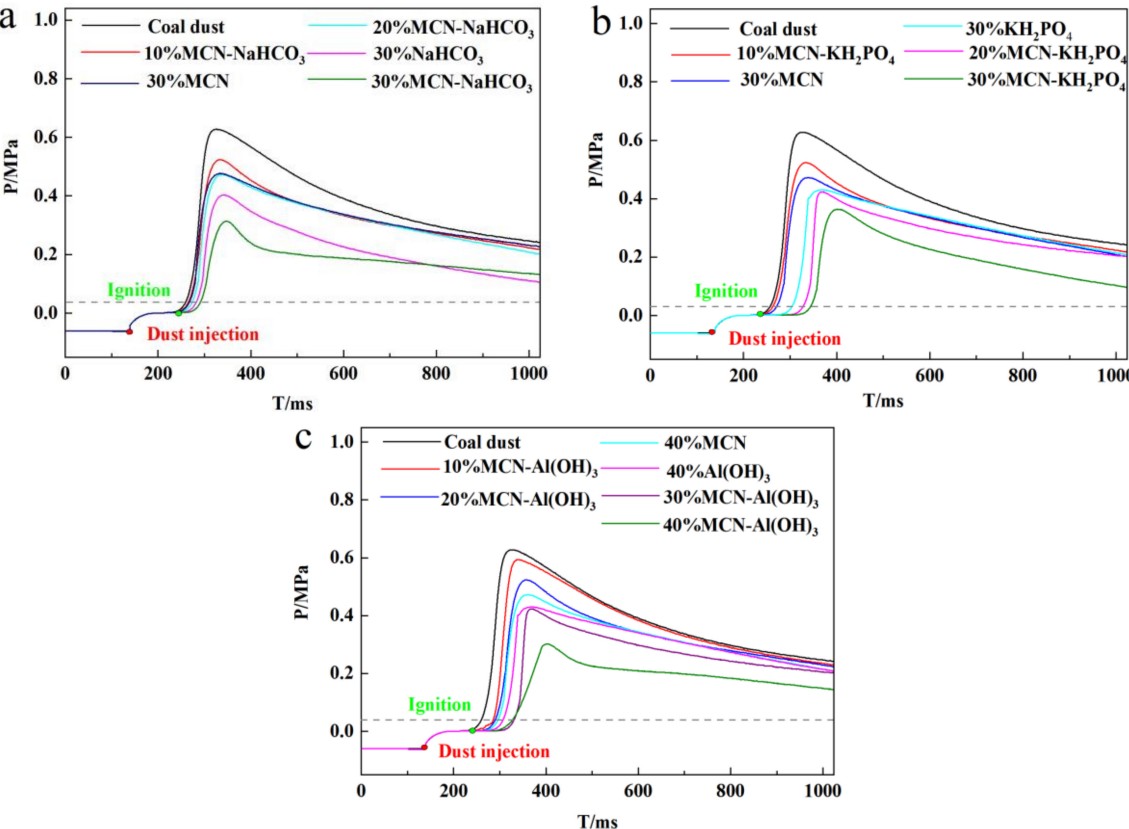

**Figure 7.** Explosion suppression characteristics of red mud based composite powder explosion inhibitors on coal dust. (**a**). MCN-NaHCO$_3$ composite powder explosion inhibitor; (**b**). MCN-KH$_2$PO$_4$ composite powder explosion inhibitor; (**c**). MCN-Al(OH)$_3$ composite powder explosion inhibitor.

In the process of explosion, the active powder of the new explosion inhibitors can play the role of pyrolysis and cooling, and the modified solid waste as the carrier can produce fine particles to quench the flame after explosion; therefore, it can play the role of explosion suppression. Considering both the explosion suppression and explosion propagation suppression results, by coal dust explosion suppression efficiency, the three kinds of red mud-based composite powder explosion inhibitors are sorted as MCN-NaHCO$_3$ composite powder explosion inhibitor > MCN-KH$_2$PO$_4$ composite powder explosion inhibitor > MCN-Al(OH)$_3$ composite powder explosion inhibitor.

Figure 8 compares the coal dust explosion suppression characteristic data between industrial solid waste-based composite powder explosion inhibitors and pure powers at the blend ratio of 30%.

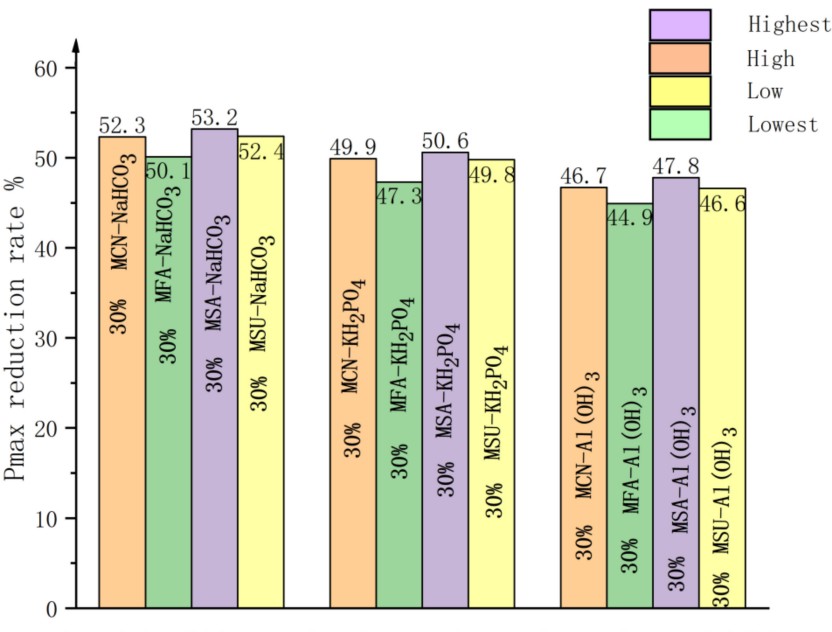

**Figure 8.** Explosion suppression characteristic data of coal dust by industrial solid waste based composite powder explosion inhibitors.

Through statistical analysis, it can be obtained from the analysis of different carrier composite powder explosion inhibitors that among a variety of industrial solid waste-based composite powder explosion inhibitors, the MSA-carried composite powder explosion inhibitor has the best explosion inhibition effect on pulverized coal. At the content of 30%, the *P*max reduction rate reaches 52%. As the carrier MSA pores are relatively developed, more loaded bodies can be combined in the compounding process, and the content of hydroxide is higher; therefore, the explosion suppression effect of MSA-carried composite powder explosion inhibitor is the best. The inhibition rate of the 30% MCN-carried composite powder explosion inhibitor on the *P*max of pulverized coal explosion reaches 49.4%. Due to the low content of hydroxide in MFA and the poor pore development in the four carriers, the explosion suppression effect of MFA-carried composite powder explosion inhibitor is the worst, and the inhibition rate of the *P*max of pulverized coal explosion is only 46.5%. Combined with the analysis results of the loading rate of the modified industrial solid waste material on the active powder material, as shown in Figure 7, it can be obtained that the explosion suppression effect of the industrial solid waste-based composite powder explosion inhibitors is related to the pore development degree of the negative carrier. The more developed the pores are, the higher the loading rate on the active powder material is, and the better the explosion suppression effect is.

Due to the different inhibition effects of active powder materials on coal dust explosion, the inhibition effects of different loaded composite powder explosion inhibitors on coal dust explosion are also different. According to the statistical results of experimental data, it is found that the composite powder explosion inhibitors are sorted as industrial solid waste-based $NaHCO_3$ composite powder explosion inhibitor > industrial solid waste-based $KH_2PO_4$ composite powder explosion inhibitor > industrial solid waste-based $Al(OH)_3$ composite powder explosion inhibitor. It is concluded that the active powder material ($NaHCO_3$) has the best inhibitory effect on coal dust explosion, followed by $KH_2PO_4$, and then $Al(OH)_3$.

### 3.2. Cost of Industrial Solid Waste-Based Composite Powder Explosion Inhibitors

The costs of the industrial solid waste-based composite powder explosion inhibitors under different preparation methods were calculated and compared with conventional powder inhibitors over economy and efficiency. Table 5 gives the costs of consumables,

namely, the price paid for purchasing the laboratory chemicals, used for preparing the industrial solid waste-based composite powder explosion inhibitors, without considering the expenses related to the material analysis and the experimental apparatuses. In addition, Table 5 gives the average amount of consumables used for preparing a kilogram of the industrial solid waste-based composite powder explosion inhibitors using the WCSC and DCBM methods.

**Table 5.** Average consumption of 1 kg of modified solid waste.

|  | Consumable | Average Consumption | Unit Price (RMB) | Mass or Volume |
|---|---|---|---|---|
| Average consumption of 1 kg of modified solid waste | Absolute ethanol ($C_2H_6O$) | 0.3 L | 5 | 500 mL |
|  | Distilled water | 4 L | 10 | 40 L |
|  | Ammonia water | 0.4 L | 7 | 500 mL |
|  | Hydrochloric acid | 0.1 L | 10 | 500 mL |
| The average amount of consumables used in the preparation of 1 kg of industrial solid waste based composite powder explosion inhibitors | $C_2H_6O$ | 0.3 L | 5 | 500 mL |
|  | Distilled water | 6 L | 10 | 40 L |
|  | $NaHCO_3$ | 0.5 kg | 6 | 500 g |
|  | $KH_2PO_4$ | 0.5 kg | 10 | 500 g |
|  | $Al(OH)_3$ | 0.5 kg | 8 | 500 g |

Table 5 gives the average amount of consumables used for modifying each kilogram of industrial solid waste. The calculations show that it costs an average of RMB11.6 to modify a kilogram of industrial solid waste.

In the case of the WCSC method, the consumables used include modified industrial solid waste, carrier, ethanol and distilled water. In the case of the DCBM method, only a modified industrial solid waste and a carrier are needed. Figure 9 gives the costs of each kilogram of the industrial solid waste-based composite powder explosion inhibitors.

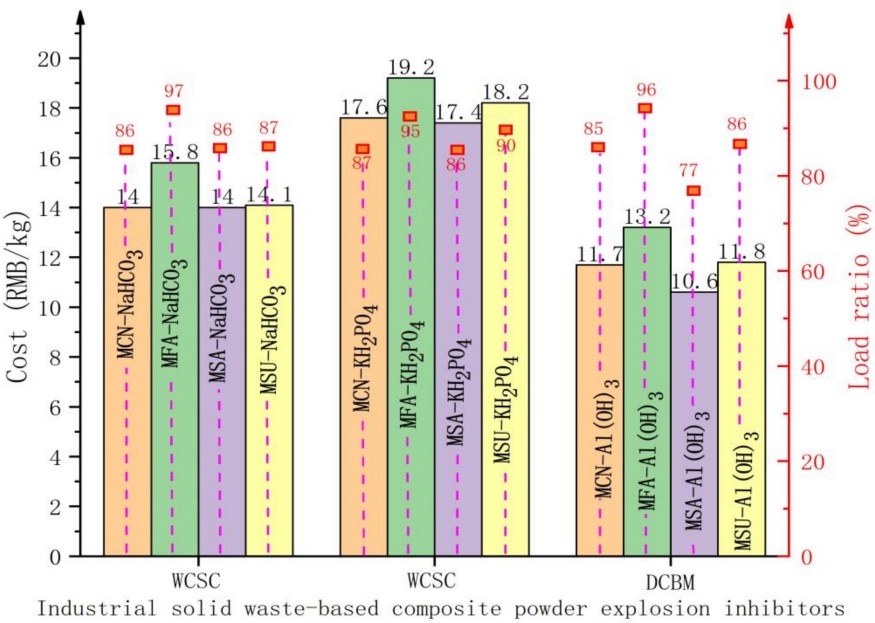

**Figure 9.** Cost of different types of industrial solid waste based composite powderexplosion inhibitors.

As calculated, when the WCSC method is used, the average cost is RMB14.4/kg for the industrial solid waste-based $NaHCO_3$ composite powder explosion inhibitor and RMB18.1/kg for the industrial solid waste-based $KH_2PO_4$ composite powder explosion inhibitor; when the DCBM method is used, the average cost is RMB11.8/kg for the industrial solid waste-based $Al(OH)_3$ composite powder explosion inhibitor and RMB14.9/kg for the industrial solid waste-based composite powder explosion inhibitor. By comparison,

while WCSC method is more costly than the DCBM method, as it consumes more chemical reagents, its load ratio works more effectively on coal dust explosion. Hence, the industrial solid waste-based composite powder explosion inhibitors prepared using the WCSC method is more efficient in suppressing coal dust explosion than those prepared using the DCBM method.

In the ZYBG-12Y mine intrinsically safe pipe explosion suppressing system (China Coal Technology & Engineering Group, Chongqing province), one of the most popular coal dust explosion units for industrial production in China, ABC dry powder is used as the powder suppressor. Under the Chinese national standard, all ABC powder inhibitors must contain a minimum of 50% $(NH_4)H_2PO_4$ [29]. Studies have demonstrated that $(NH_4)H_2PO_4$ has a better explosion suppression performance than ABC dry powder [30]. Some researchers [21,31] investigated the coal dust explosion suppression property of $(NH_4)H_2PO_4$. The maximum coal dust explosion pressure is 0.58 MPa; $(NH_4)H_2PO_4$ can bring the maximum coal dust explosion pressure down to 0.1 MPa at the blend ratio of $\Phi = 80\%$. The experiment revealed that the MCN-NaHCO$_3$ composite powder explosion inhibitor and the MCN-KH$_2$PO$_4$ composite powder explosion inhibitor can effectively suppress coal dust explosion at the blend ratio of 40%, while the MCN-Al(OH)$_3$ composite powder explosion inhibitor can do so at the blend ratio of 50%.

As discussed, by explosion suppression performance, the industrial solid waste-based composite powder explosion inhibitors are sorted as $(NH_4)H_2PO_4$ suppressor > ABC dry powder suppressor. For 1 kg of coal dust, if the MCN-NaHCO$_3$ composite powder explosion inhibitor is used, 0.4 kg is enough to fully suppress a coal dust explosion; hence, the explosion suppression cost would be RMB6.32; if ABC dry powder is used, 0.8 kg is needed to attain the same result. At the market price of RMB5/kg for ABC dry powder, the explosion suppression cost would be RMB4/kg. In the case of the MCN-NaHCO$_3$ composite powder explosion inhibitor, the amount needed is about a half that of the ABC dry powder suppressor and the cost is about RMB2.32/kg higher. The amounts and costs of the other industrial solid waste-based composite powder explosion inhibitors are just comparable to those of the MCN-NaHCO$_3$ composite powder explosion inhibitor ($\pm$10–12%).

As the explosion suppression costs of the industrial solid waste-based powder inhibitors indicated herein are laboratory costs, the industrial-scale application will be much less costly. In addition, the government also grants tax reliefs on industrial solid waste-based products, which will further reduce the application cost of industrial solid waste-based inhibitors. Furthermore, as these composite powder explosion inhibitors are consumed at only a half of the amount needed when ABC dry powder is used, the capacity of the suppressing system, and consequently, the manufacturing cost of the suppressing unit, will be reduced remarkably.

In summary, using industrial solid waste-based composite powder explosion inhibitors to suppress a coal dust explosion can not only effectively suppress the use cost of inhibitors and the manufacturing cost of the suppressing units, but also enhance the coal dust explosion suppression efficiency. As far as the suppression efficiency and suppression cost are concerned, industrial solid waste-based composite powder explosion inhibitors have good application prospects.

## 4. Conclusions

Based on the consideration of environmental protection and the underground safety of coal mine, a preparation method of new composite powder explosion inhibitor based on industrial solid waste is proposed to suppress an underground coal dust explosion, and the explosion suppression effect of coal dust is experimentally studied. It is found that modified solid waste materials—modified red mud (MRM), modified slag (MSA), modified fly ash (MFA) and modified sludge (MSU)—as the load, NaHCO$_3$, KH$_2$PO$_4$ and Al(OH)$_3$ as the carrier, and the industrial solid waste-based composite powder explosion inhibitors

can be successfully compounded by wet coating by solvent–crystallization (WCSC) and dry coating by ball milling (DCBM).

In terms of the underground safety of a coal mine, the new composite powder explosion inhibitors can effectively reduce the explosion overpressure of coal dust and suppress the explosion of pulverized coal at the content of 40–50%. The paper also compares the explosion suppression effect and economic analysis between the explosion suppression experimental results and the powder explosion inhibitor commonly used in industrial production. It is found that using industrial solid waste-based composite powder explosion inhibitors to suppress a pulverized coal explosion can effectively reduce the use cost of an explosion inhibitor and the manufacturing cost of explosion suppression equipment and increase the explosion suppression effect on coal dust. In terms of the explosion suppression effect and cost, industrial solid waste-based composite powder explosion inhibitors have a good application prospect. In addition, considering the environmental protection of coal mines, the modified industrial solid waste combined with a small amount of active explosion suppression materials can synthesize explosion inhibitors. The modified solid waste has no heavy metals or toxic substances, and a small amount of $NaHCO_3$, $KH_2PO_4$ and $Al(OH)_3$ basically have no impact on the underground soil environment of coal mines. Therefore, the industrial solid waste-based composite powder explosion inhibitor can avoid polluting the application environment.

According to the current explosion suppression measures in the coal mine, the industrial solid waste-based composite powder explosion inhibitor can be sprinkled into the roadway with more coal dust in a certain proportion. When the coal dust is lifted by the shock wave, the new composite powder explosion inhibitor can be mixed in the dust cloud to prevent and suppress the coal dust explosion. In addition, the industrial solid waste-based composite powder explosion inhibitor can be placed in the explosion-proof groove at the top of the roadway. When the shock wave overturns the explosion-proof groove, the new composite powder explosion inhibitor forms a dust cloud, which can suppress the explosion flame. In addition, the industrial solid waste-based composite powder explosion inhibitor can be placed in the fixed explosion inhibitor. When the explosion suppression device is triggered, the new composite powder explosion inhibitor is ejected within the specified time, forming a dust cloud in the roadway, so as to achieve the purpose of suppressing the coal dust explosion.

**Author Contributions:** Conceptualization, B.L. and Y.Z. (Yuyuan Zhang); data curation, B.L. and N.M.; formal analysis, B.L.; investigation, B.L. and K.X.; methodology, B.L., K.X. and Y.Z. (Yansong Zhang); supervision, Z.H. and N.M.; validation, K.X. and Z.H.; writing—original draft, K.X.; writing—review and editing, B.L., K.X., Y.Z. (Yuyuan Zhang) and Y.Z. (Yansong Zhang). All authors have read and agreed to the published version of the manuscript.

**Funding:** This work was supported by the National Natural Science Foundation of China (52074066).

**Institutional Review Board Statement:** Not applicable.

**Informed Consent Statement:** Informed consent was obtained from all subjects involved in the study.

**Data Availability Statement:** Not applicable.

**Conflicts of Interest:** The authors declare no conflict of interest.

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
