# Peer review of "Study on a New Type of Composite Powder Explosion Inhibitor Used to Suppress Underground Coal Dust Explosion"

_applsci, doi:10.3390/app11188512_

Round 1

Reviewer 1 Report

The article is suitable for publication 

Author Response

Thank you for your support for my paper. Your support will be my motivation.

Reviewer 2 Report

The manuscript entitled Study on a New Type of Composite Powder Explosion Inhibitor Used to Suppress Underground Coal Dust Explosion elucidated the consideration of environmental protection and underground safety of coal mine. Details, a preparation method of new composite powder explosion inhibitor is proposed to suppress underground coal dust explosion, and the explosion suppression effect of coal dust is studied. I think there are still some issues that need to be corrected. I would recommend the manuscript for major revision
Here are comments for the authors:

Introduction must be improved by incorporating more recent references and findings on the topic and by doing a comparison. 
Importantly, It would be nice, if authors would show a schematic representation of the topics coverd to improve the impact of this manuscript. Please address concise and clear scheme in a figure.

Table 2 should be removed and please describes sentences in your manuscript.

Figure 2 and Figure 3 should be merged and more detailed in your manuscript. 
Table 4 should be removed and please describes sentences in your manuscript.
L 266, please put eqn no. 
Figure 7 should be detailed and please describes sentences in your manuscript. In general, the materials in the mixer have different physical and chemical properties. The authors should be provided materials' properties and safety in modification and compounding process.
Table 7 and Table  8 should be merged and more detailed in your manuscript.

In conclusion, please the contents detailed should be addressed including future scope and applications in the consideration of environmental protection and underground safety of coal mine in compounding process aspects.The authors need to provide more discussion on why new composite powder explosion inhibitor is scientifically significant. What applications can it be used and how will its use in these applications improve the state of the art?

Reviewer 3 Report

The paper idea and topic are very important and interesting. However, the manuscript features some shortcomings, particularly regarding technical issues. Therefore, the paper cannot be accepted in the present form and moderate revision is required.

It looks like the text was done in a hurry. It is necessary to read and correct the text in detail because currently some parts of the text look like a copied parts from another article because capital letters, comas and dots appear somewhere where they do not belong. Generally, some sentences are too short as telegraphic, and some are confusing.

Abstract should be slightly reworked. The authors should keep only the sentences about the real overarching goals and main objectives as well as the purpose of the study. Excessive detail is unnecessary.

Some of technical shortcomings…

line 107: "...Table 1 .." => "...Table 1."

line 147: " ...following reasons (...)." =>" ...following reasons (see ...) :"

line 174 “As shown in Fig. 2,  100 mg MSA was added…” => “100 mg MSA was added … (see Fig. 2)”

Figure 7 is taken from the authors recent paper Liu, B., Xu, K., Zhang, Y., Ge, J., & Zhang, Y. (2021). doi 10.1016/j.jlp.2021.104619. Journal of Loss Prevention in the Process Industries, 104619. It should be noted in the caption of Figure 7.

Figure 8 should be enlarged and/or text on axes.

line 371: Match?

Obviously, the authors did not use the template so the Sections Author Contributions, Data Availability Statement, Conflicts of Interest are missing.

The format of citations should be numeric. All references should be written in the same way, according to the journal style. Please correct.

Round 2

Reviewer 2 Report

The authors addressed all concerns raised and improved the manuscript for the publication.

This manuscript is a resubmission of an earlier submission. The following is a list of the peer review reports and author responses from that submission.

Round 1

Reviewer 1 Report

The authors presented the results of research related to experimental study on the feasibility of preparing composite powder explosion inhibitors from industrial solid wastes. And the next step was compare their effectivness on coal dust explosion.

The manuscript sent for evaluation consists of 26 pages, including 22 pages of text with tables and figures, and 4 pages for references. References contain 28 bibliographic items.

After reading the text, in general, I assess the scientific quality of the article as suitable for publication, but after major revision.

Abstract is generally correct and contains all the necessary information, such as methods, results, main statements. However, he has more sentences than usual.

The introduction is extensive and generally in line with the topic of the publication. There is clear description indicating a lack of knowledge in the topic under consideration and is justification for undertaking the research. But, article does not write goal (or aim). I think, that is very important for clear presentation experimental samples, methods, results and conclusion.

The formal arrangement does not correspond to the content of the scientific article. Authors should have a clear scientific content: introduction, motodology (description and preparation of samples + description of methods used). The results and discussion are a presentation of the obtained results and summary obtained results as conclusion.

The methodology is written with an explanation of the methodological procedure of the experiment.

I have not assessed the correctness of the language as English is not my mother tongue.

Please, find below s remarks, which, I hope, make your manuscript better:

- what does Qty means? (Table 7 and Table 8)

- it is important to give a name for each formula (eg C2H6O)

Reviewer 2 Report

The paper written by the following Authors: Bo Liua, Yuyuan Zhanga, Kaili Xua, Yansing Zhangc, Zheng Haoa, Ning Ma, entitled “Study on a New Type of Composite Powder Explosion Inhibitor Used to Suppress Underground Coal Dust Explosion” presents an interesting study on an experimental study on the feasibility of preparing composite powder explosion inhibitors from industrial solid wastes.

Although the paper is interesting, I have some major concerns:

Title

The title reflects the results presented here.

Abstract

The abstract is lacking the aim of the material and methods description as well as an informative conclusion. It should be written in more details.

There is no Material and Methods chapter and results. Authors connected description of study preparation together with results. It is difficult to understand the core of the study.

Moreover, there is no discussion chapter. Authors should refer their results to the existing literature.

There is no information about the experimental system applied for this study. It should be included in the manuscript.

Figures:

Figure 8 the font size is too small.

Round 2

Reviewer 1 Report

The authors incorporated the requested comments. I agree with its publication.

Author Response

Thank you again for your guidance on this paper. Your opinions are of great help to my academic research.

Reviewer 2 Report

Discussion part still needs to be improved. The Authors did not discuss their results with the latest findings.

Round 3

Reviewer 2 Report

Discussion part still needs to be improved.

Still the Authors did not discuss their results with the latest findings.

Still only one reference was applied.